# Recent Advances in the Synthesis of Marine-Derived Alkaloids via Enzymatic Reactions

**DOI:** 10.3390/md20060368

**Published:** 2022-05-30

**Authors:** Bi-Shuang Chen, Di Zhang, Fayene Zeferino Ribeiro de Souza, Lan Liu

**Affiliations:** 1School of Marine Sciences, Sun Yat-Sen University, Zhuhai 519082, China; chenbsh23@mail.sysu.edu.cn (B.-S.C.); zhangd67@mail2.sysu.edu.cn (D.Z.); cesllan@mail.sysu.edu.cn (L.L.); 2Southern Marine Science and Engineering Guangdong Laboratory (Zhuhai), Zhuhai 519082, China; 3Centro Universitário Planalto do Distrito Federal, Universidade Virtual do Estado de São Paulo (UNIPLAN), Campus Bauru 17014-350, Brazil

**Keywords:** alkaloids, natural product, synthesis, biotechnological approaches, enzymatic reactions, chemoenzymatic strategies

## Abstract

Alkaloids are a large and structurally diverse group of marine-derived natural products. Most marine-derived alkaloids are biologically active and show promising applications in modern (agro)chemical, pharmaceutical, and fine chemical industries. Different approaches have been established to access these marine-derived alkaloids. Among these employed methods, biotechnological approaches, namely, (chemo)enzymatic synthesis, have significant potential for playing a central role in alkaloid production on an industrial scale. In this review, we discuss research progress on marine-derived alkaloid synthesis via enzymatic reactions and note the advantages and disadvantages of their applications for industrial production, as well as green chemistry for marine natural product research.

## 1. Introduction

Seas and oceans account for more than 70% of Earth’s surface, and their resources occupy 80% of global biomass, in terms of animals, plants, and microorganisms [1]. Unique marine environments include the deep sea, Arctic and Antarctic zones, and symbiotic environments, making marine-derived organisms significantly different from terrestrial organisms. Thus, it would be reasonable to presume that marine organisms have diversified structures and functions [2]. Most marine organisms are considered an important resource for pharmaceutical and other economically useful applications [3,4,5]. In recent years, the exploitation of marine natural products, such as the secondary metabolites of marine organisms, has tremendously increased due to their structural uniqueness, excellent activity, and environmentally friendly characteristics [6,7]. As a result, a surprisingly large number of marine natural products with novel structures have been discovered, due to the expansion of relevant research and the rapid growth of marine organism chemistry over the last few decades [8,9]. These products include polyketides, terpenes, alkaloids, and peptides.

Among the many classes of marine natural products, alkaloids represent a large and structurally diverse group of secondary metabolites found in marine environments [10,11]. Of note, the term alkaloid is not a universally accepted definition; however, it comprises an extensive and important group of nitrogen-containing compounds, including amides, nitro, and nitroso compounds. The primary metabolites of amino acids, proteins, and porphyrins are excluded from the class of alkaloids [11]. Approximately 40,000 compounds have been described as alkaloids under the broad term [12,13,14], and about 2500 compounds classified as alkaloids have been isolated from marine organisms and microorganisms, according to research published between 2000 and 2021 [9,15,16,17,18,19,20,21,22,23,24,25]. The extensive chemodiversity of alkaloids produced by marine organisms/microorganisms has attracted significant attention due to its important pharmacological activity [26,27,28,29]. Notably, many of the known marine-derived alkaloids have long been recognized as pharmacologically interesting agents and have a long history of therapeutic applications. For example, norditerpenoid alkaloids exhibit strong pharmacological effects, and their use as insecticides has remained for a long period [26]. Consequently, alkaloids also have other economically useful applications such as fine chemicals, cosmetics, and functional personal-care products.

Because alkaloids play an important role in both traditional and modern medicine, the development of their production routes has received significant interest in preparative organic chemistry. Remarkably, numerous alkaloids have been produced that exert prominent commercial value, with an annual output of hundreds of thousands of tons [11]. For the preparation of marine-derived alkaloids, isolation from their marine source is still the most common approach, as summarized by the existing literature [9,15,16,17,18,19,20,21,22,23,24,25]. The isolation of marine-derived alkaloids from their natural source has its benefits. For example, alkaloids for public demand isolated from marine microorganisms are considered to be a sustainable development model. However, many alkaloids are isolated from their marine source with typically low yields in unit values of μg/L; thus, they cannot meet industrial demand. Specifically, even if traditional isolation was successfully employed on large scale, it is typically a very expensive and time-consuming process. Moreover, obtaining alkaloids by direct isolation from the marine source has obvious limitations. For instance, some marine-derived organisms cannot be reproduced under laboratory conditions, resulting in unpredictable supply situations. As an example, (−)-sparteine, a type of quinolizidine alkaloid isolated from marine sources, can be used as a chiral ligand; however, it is quite scarce at the moment [30].

Alternatively, many marine-derived alkaloids can be prepared on an industrial scale using organic synthetic strategies. Traditional synthetic methods frequently involve small molecules as reagents, resulting in long sequences that involve extensive protective groups, typically low overall yields, and the significant generation of waste [31]. Modern synthetic methods typically use catalysts based on precious metals, which dramatically limits the overall economic benefits of the proposed alkaloid synthesis methods [32,33]. Today, this situation has changed with the development of biocatalysis [34]. Specifically, biocatalysis denotes the use of enzymes for marine-derived alkaloid preparation for research and industrial scales. Enzymes possess significant advantages over most traditional chemical catalysts, as they are biodegradable, can be used under mild conditions, and generally show excellent selectivity. Because the use of enzymes in many synthetic routes has provided significant opportunities to develop novel strategies for both in vitro and in vivo alkaloid production, an increasing number of research studies have been published related to the biosynthesis of alkaloids or chemoenzymatic alkaloid synthesis.

Because only a few reviews have been published on the synthesis of marine natural products via enzymatic reactions [10,11], we determined that it would be of significant benefit to present a discussion focusing on enzymatic methodology and the range of marine-derived alkaloid synthesis applications. Therefore, we limited our discussion to the current literature (from 2005 to 2022) for different applications of biocatalysis in the asymmetric synthesis of alkaloids, and recent breakthroughs in chemoenzymatic alkaloid synthesis were highlighted.

## 2. Marine-Derived Alkaloids

There are yearly reviews entitled “marine natural products” published in the journal Natural Product Reports, which refer to compounds isolated from marine sources [15,16,17,18,19,20,21,22,23,24,25,27]. These reports show that the number of reported marine-derived alkaloids increased rapidly from 64 in 2016 to 160 in 2021. Although there are several proposed definitions of the term ‘alkaloid’, the most acceptable one is introduced by S. William Pelletier “An Alkaloid is a cyclic organic compound containing nitrogen in a negative oxidation state which is of limited distribution among living organisms” [10,35]. According to this broad definition, this class of natural compounds can by classified into seven categories, including indole alkaloids, carboline alkaloids, guanidine alkaloids, bromotyrosine alkaloids, pyrrole alkaloids, diterpene alkaloids, and other alkaloids. For example, one report summarized 149 marine alkaloids from sponges on the basis of their structures, from 2015 to 2020, and revealed that the seven different categories contained 26, 12, 30, 16, 30, 18, and 17 alkaloids [35]. The rise in the study of marine sources has extended to the spectrum of marine organisms for alkaloid isolation. Marine-sourced bacteria, fungi, and sponges continue to be a predominant source of new alkaloids; however, few alkaloids have been isolated from cyanobacteria, dinoflagellates, algae, echinoderms, cnidarians, bryozoans, mollusks, tunicates, and mangrove, which are typically common sources for natural product discovery [36,37,38].

Marine-derived alkaloids possess potent pharmacological activity. For example, one imidazole alkaloid, paenidigyamycin A, which was isolated from Ghanaian *Paenibacillus* sp. DE2SH showed significantly high antiparasitic activity of IC_50_ 0.75 µM, making it just as effective as amphotericin B (IC_50_ 0.31 µM) [39]. Shang et al. showed that C_19_-aconitine diterpenoid alkaloids, isolated from *Aconitum carmichaelii*, had a significant effect on the analgesic activity of mice when administered in acetic acid at a dose of 0.5 mg/kg of body weight [40,41]. C_20_-vakognavine diterpenoid alkaloids have been shown to exert noticeable anti-inflammatory activity nearly equal to aspirin; thus, they may be a potential aspirin alternative [42,43]. Angustilongines have shown strong anticancer cytotoxicity against various human cancer cell lines with excellent IC_50_ values ranging from 0.02 to 9.0 µM [44,45].

## 3. Enzymatic and Chemoenzymatic Approaches for Alkaloids

The significance of alkaloids has resulted in the search for strategies to obtain these compounds. Herein, we describe how some enzymatic and chemoenzymatic reactions can produce alkaloids. Briefly, we provide the information on the synthesis of 18 different kinds of marine-derived alkaloids via enzymatic and chemoenzymatic approaches, as well as the pros and cons of their application for industrial-scale production.

The chemoenzymatic synthesis of cryptophycins and arenastatins was described [46] by the cloning, expression, and use of cryptophycin thioesterase (CrpTE), combined with the synthesis of seco-cryptophycin and seco-arenastatin *N*-acetylcysteamine thioester substrates. Cryptophycins and arenastatins are secondary metabolites that were isolated from marine sponge *Dysidea arenaria* by Kobayashi and Kitigawa [47]. The cyclization of unit A derivates was obtained by the enzymatic modification of the ring closure sites, the cyclization of unit C derivates, and the *N*-acetylcysteamine thioester substrates, which were modified at the C-6 position. This method investigated CrpTE in the structural analogues of units A and C for the production of cryptophycin and arenastatin. The marine-derived alkaloids cryptophycins and arenastatins were obtained in five steps with total yields of ~15%. The hydrolysis activity of Crp TE for chemoenzymatic synthesis provided a new and efficient route for the cryptophycin/arenastatin natural products and the synthesis of this important class of antitumor agents.

(−)-Gephyrotoxin-223 (indolizidine 223 AB) (Figure 1) is one of the 3,5-disubstituted indolizidines and belongs to a large group of simple bicyclic alkaloids [48]. Ghodsi et al. performed the formal asymmetric synthesis of (−)-gephyrotoxin-223 via enzymatic desymmetrization [49]. The key synthesis intermediate was chiral synthon [(+)-(2*R*,6*S*)-monoacetate/(−)-(2*S*,6*R*)-monoacetate] (**1**), which was prepared by enzymatic hydrolysis and enzymatic acetylation in the presence of *Aspergillus niger* lipase (ANL). Using chiral synthon as the starting material, the alkaloid ((−)-gephyrotoxin-223) was achieved in good yield. Of note, the chiral synthon (**1**) obtained by the enzymatic reactions could be used as a chiral building block in the enantioselective synthesis of a series of piperidine alkaloids (Figure 1).

Marine-derived fungi of genera *Claviceps*, *Penicillium*, and *Aspergillus* will produce different secondary metabolites, including fumigaclavine C (**5**) (Figure 1) [50,51], an ergot alkaloid of the clavine type. All ergot alkaloids possess diverse biological and pharmacological activities. Steffan et al. performed chemoenzymatic synthesis of prenylated indole derivatives, with fumigaclavine C (**5**) as the best example, by using 4-dimethylallyltryptophan synthase from *Aspergillus fumigatus* (Figure 1). In addition, 4-prenylated indole derivatives and 7-prenylated indole derivatives were produced, using aromatic prenyltransferases, by FgaPT2 and DMATS as biocatalysts, respectively. Significantly, to conduct these chemoenzymatic reactions, the authors also used 10 different substrates from tryptophan derivatives and tryptophan-containing cyclic dipeptides, and different conversion rates were obtained [52,53,54].

(*S*)-Norcoclaurine (**9**) (Figure 2) is a benzylisoquinolinealkaloid. Benzylisoquinoline alkaloids possess diverse biological activities and are widely employed as pharmaceuticals. In fact, (*S*)-norcoclaurine is currently considered one of the best β1-adrenergic drugs in its class, with a pharmacokinetic profile that favorably competes with dobutamine [55]. Because dobutamine is currently marketed as a racemic mixture, it is particularly important for achieving enantioselective (*S*)-norcoclaurine. Nevertheless, the availability of natural (*S*)-norcoclaurine from marine/terrestrial sources is still very limited; thus, the development of a synthetic strategy using cheaper and cleaner synthetic routes is urgently needed. Bonamore et al. developed an efficient, stereoselective, green synthetic route for (*S*)-norcoclaurine using the recombinant (*S*)-norcoclaurine synthase (NCS) enzyme (Figure 2). Inexpensive tyrosine (**6**) and dopamine (**8**) were used as the starting materials to prepare (*S*)-norcoclaurine using the green Pictet–Spengler reaction. First, tyrosine underwent oxidative decarboxylation to produce 4-hydroxyphenylacetadehyde (**7**) [56]. Then, dopamine and NCS with ascorbate were added to avoid oxidation and to afford the desired (*S*)-norcoclaurine using a one-pot, two-step process. Under the optimized reaction conditions, (*S*)-norcoclaurine was obtained with a yield greater than 80% and with an *ee* value of 93%. Furthermore, the enzyme could be recycled and maintain its excellent activity and stability. These results represented an example of a chemoenzymatic synthetic strategy for (*S*)-norcoclaurine production, which could open up a novel, efficient, and green strategy for benzylisoquinoline alkaloid production.

Prenylated indole alkaloids are natural products that are mainly produced by marine organisms, especially in the genera *Penicillium* and *Aspergillus* of ascomycota. For example, brevicompanine G was isolated from deep ocean sediment-derived *Penicillium* sp. [57], representing a large group of compounds that contain indole and isoprenoid moieties with diverse biological and pharmaceutical activity. A number of prenylated indole alkaloids or/and their derivatives have been found to exhibit cytotoxicity. For example, tryprostatins A and B (Figure 2), as well as their diastereomers, have shown better cytotoxicity against various cancer cell lines than etoposide [58,59,60]. In addition, notoamide A (Figure 2) was found to exhibit moderate cytotoxicity toward HeLa and L1210 cells [61], and tryprostatins A and fumitremorgin C (Figure 2) were reported to be active inhibitors of the ABC transporter and breast cancer resistance protein, respectively [62]. Moreover, both compounds could reverse the resistance of some tumor cell lines [63]. Hinnuliquinone and semicochliodinols A and B were also found to inhibit HIV-1 protease [64]. Over the past decade, the biosynthesis of prenylated indole alkaloids has been well demonstrated by molecular biological and biochemical characterization, as summarized by several excellent reviews [65,66]. Significant progress has also been achieved in the chemoenzymatic synthesis of prenylated indole alkaloids. Although many prenylated indole alkaloids are derivatives of the same amino acids, the differences in prenylation position and manner represent large structural diversity within the prenylated indole alkaloids. Thus, the key steps for the chemoenzymatic synthesis of prenylated indole alkaloids have focused on the prenyl transfer reactions catalyzed by prenyltransferases. Figure 3 shows one example of chemoenzymatic synthesis of prenylated simple indole derivatives (**10**, **12**, **13**, **14**) using prenyltransferases. The same indole derivate (**11**) was used as a substrate for mono-prenylation using prenyltransferases FtmPT1 or CdpNPT, FgaPT2, and 7-DMATS. Diprenylated indole was produced using FgaPT2, and 7-DMATS by tandem incubation, while cyclic dipeptide containing one tryptophan was used as the substrate to produce prenylated derivatives using CdpNPT, AnaPT, FgaPT2, and 7-DMATS [67]. Notably, many derivatives with different indole positions have been obtained using the abovementioned strategy [68].

A group of cyanobacterial depsipeptides called cryptophycins has been found to exhibit significant activity against drug-resistant tumors [69]. Although cryptophycins have shown to be promising potent anticancer agents, their synthesis (semi-synthesis, total synthesis, or chemoenzymatic synthesis) is still a challenge. Thus, synthetic strategies are important research topics in the field of marine natural products [70]. Significant effort has been made to produce suitable amounts of desirable target compounds and new related analogues with modified biological functionality. In one example, Ding et al. identified the cryptophycin gene cluster, where the gene cluster consisted of polyketide synthase (PKS), non-ribosomal peptide synthetase (NRPS), and tailoring enzyme genes. Multifunctional enzyme CrpD-M2 was identified as a candidate gene for the enzymatic chemoenzymatic synthesis reactions of cryptophycins (**19** to **27**) through molecular biological and biochemical characterization (Figure 4). CrpD-M2 was found to be composed of condensation–adenylation–ketoreduction–thiolation (C–A–KR–T) domains. Thus, the cryptophycins were chemoenzymatically synthesized using ester bond-forming non-ribosomal peptide synthetase (CrpD-M2) and Crpthioesterase in four steps (Figure 4A–D). The ketoreduction domain of CrpD-M2 using coenzymes NADPH and NADH resulted in 2-hydroxy acid (**18**); subsequently, the product was macrolactonized by Crpthioesterase [71]. It is expected that analogues with modified physicochemical functionality will be produced using the same strategy with CrpD-M2 as the chemoenzymatic reagent.

Many of the marine natural products commonly used in modern medicine are chiral compounds. Because enantiomerically pure compounds have received significant interest in academia and industry, the development of synthetic strategies for optically pure chiral natural products has become an important topic in preparative organic chemistry. One common way to prepare chiral compounds in an enantiomerically pure form involves the conventional kinetic resolution of racemic compounds using chemical catalysts or lipases [72]. However, using this strategy, the desired enantiomer could be obtained with a maximum theoretical yield of 50%, which would significantly reduce the efficiency of this process, especially in situations where the starting materials were derived from natural sources, or where their synthesis required a multistep route. Significant progress has been made in the application of biocatalysts (except for lipases) for the synthesis of chiral amines based on deracemization. For example, transaminase, ammonia lyase, amine dehydrogenase, imine reductase, and amine oxidase have been used for the industrial synthesis of chiral amine [73]. As amines, marine-derived alkaloid natural products typically exhibit interesting biological activities; thus, the application of biocatalysts for the deracemization of racemic alkaloids has received significant attention to afford enantiomerically pure compounds. For example, Ghislieri et al. demonstrated the use of monoamine oxidase variants from *Aspergillus niger* (MAO-N D9) for the deracemization of marine-derived (±)-eleagnine (**28**) and (±)-leptaflorine (**29**) to their corresponding enantiomerically alkaloid natural products (Figure 5) [74]. (*R*)-Eleagnine (**30**) in 99% *ee* was accumulated after repeated cycles, and it underwent the oxidation of (*S*)-coniine and nonselective chemical reduction of imine with BH_3_−NH_3_ in the following step. (*R*)-Eleagnine (**30**) was obtained with a yield of 93%, while (*R*)-leptaflorine (**31**) (in 99% *ee*) was afforded with a yield of 99% using the same strategy.

Lipowicz et al. isolated nitrile hydratase (Nhase) from the Mediterranean sponge *Aplysina cavernicola* and converted nitrile aeroplysinin-1 (**32**) to dienone amide verongiaquinol. Dienone amide verongiaquinol originated from the brominated isoxazoline alkaloids, which were secondary metabolites of the marine sponge. Therefore, these results indicated the chemoenzymatic synthesis of marine-derived alkaloids (Figure 6) [75]. *Aplysina cavernicola* uses NHase enzyme as a chemical defense, compared to other NHase compounds. Thus, the hydration of the nitrile group from aeroplysinin-1 (**32**) into dienone amide (**33**) occurred when NHase catalyzed this reaction. This enzyme not only converted nitrile into amides but also eliminated the hydroxyl group from aeroplysinin-1 (**32**). The elimination of the hydroxyl group created a double bond in the product, and the enzymes also generated an oxo function in dienone amide by demethylating the methoxy substituent. Of note, a series of compounds structurally related to aeroplysinin-1 were tested using the studied Nhase, and none were accepted by the enzyme. The fact that the tested analogues, except for aeroplysinin-1, did not bind to the studied Nhase suggested that the enzyme had very strict substrate specificity. Thus, the application of this strategy for the transformation of other closely related alkaloids may be limited. Remarkably, this synthetic strategy benefited from the high stability of the enzyme, as the optimum pH and temperature were found to be 8.0 and 41 °C, respectively. The effects of different metal ions on enzymatic activity were also described [76]. Manganese was shown to be the best ion capable of restoring enzymatic activity, while cobalt and nickel ions were relatively less effective. Zinc, copper, and iron ions had no effect on enzymatic activity.

Petrosins were first isolated by Braekman et al. from the extract of marine sponge *Petrosia seriata* [77]. Petrosins are bisquinolizidine alkaloids; therefore, they belong to the studied marine-derived alkaloids. Petrosins have been found to exhibit anti-HIV activity, as they can actively inhibit the formation of syncytinum and reverse transcriptase [78]. Further evaluation of petrosin biological activity showed that petrosins significantly prevented the HIV from binding to human cells. For this reason, the development of a synthetic strategy to provide a large amount of petrosins and the creation of new analogues with modified/improved biological activity have been important research topics in the field of organic chemistry. Toya et al. accomplished total synthesis of (−)- and (+)-petrosins with lipase-mediated desymmetrization of 1,3-diol as the key step (Figure 7) [79]. The chemoenzymatic preparation of the bisquinolizidine alkaloid, (−)- and (+)-petrosin, was achieved by using several steps. The reduction of known compound (**37**) afforded 1,3-diol (**38**), which was converted to an enantiomerically pure form after lipase-mediated desymmetrization. Protection from the remaining hydroxyl group followed by hydrolysis gave carbamate (**40**), and Swern oxidation of carbamate (**40**) gave the corresponding product (**41**) in quantitative yield. The desired ester was obtained by the diastereoselective Mannich reaction. After a sequence of organic reactions, the final products, (−)-petrosin and (+)-petrosin (**46**), were obtained in 95% *ee* and 96% *ee*, respectively. Except for lipase-mediated desymmetrization, the diastereoselective Mannich reaction was an essential step in stereocontrolling the quinolizidine ring. A series of bispiperidine derivatives were also synthesized using the same strategy.

2,5-Diketopiperazines are natural products from numerous terrestrial and marine sources. These products are structurally diverse and usually exhibit interesting biological and pharmaceutical activities, including antimicrobial, anticancer, and immunosuppressant effects [80]. Nocardioazines A-B, isolated from marine-derived *Nocardiopsis* sp. CMB-M0232 by Raju et al., represent a unique type of functionalized diannulated tryptophan 2,5-diketopiperazine natural products. *Cyclo*(L-Trp-LTrp) was shown to be the relevant intermediate to nocardioazine assembly in vivo [81]. Thus, the chemoenzymatic synthesis of *cyclo*(L-Trp-LTrp) could provide significant benefits in the synthesis of nocardioazines A–B, which were described by James et al. via molecular biological and biochemical investigations [82]. Biosynthetic genes, including CDPSs, NozA, and NcdA were identified as a cluster in the genome sequence of *Nocardiopsis* sp. The CDPSs gene (cyclodipeptide synthases) was identified as a candidate for enzymatic reactions, by comparing orthologous gene clusters within different strains. Heterologous expression of the CDPSs gene in *E. coli* afforded the purified enzyme cyclodipeptide synthases. The enzymatic reaction showed that the tryptophanyl-tRNA substrates were converted to *cyclo*(L-Trp-LTrp) catalyzed by CDPSs. Remarkably, additional molecular biological and biochemical characterization revealed that NozA and NcdA exhibited strict substrate specificity, only catalyzing *cyclo*(L-Trp-L-Trp) biosynthesis from tryptophanyl-tRNA, but not binding to other aromatic aminoacyl-tRNA substrates. These experiments showed that CDPSs, NozA, and NcdA were all capable of yielding *cyclo*(L-Trp-L-Trp). The study represented a rare example of multiple phylogenetically distinct enzymes that produced the same secondary metabolites. This process may open up an entirely new approach to the chemoenzymatic synthesis of indole alkaloid diketopiperazine.

Indole alkaloids are well-known marine-derived natural products due to their interesting biological activity and complex chemical structures. Indolic nitrones, such as avrainvillamide, waikialoid, and roquefortine L, comprise a rarely observed group of indole alkaloids, because they feature a triazaspirocyclic skeleton derived from an unusual rearrangement of the kiketopiperazine core [83]. Nevertheless, indolic nitrones possess potent properties including antibacterial, anticancer, antiparasitic, and insecticidal activities. For example, avrainvillamide was shown to exhibit promising antiproliferative activity, and the proposed mechanism was that the nitrone functional group was nucleophilically attacked by cystein residues in the cellular proteins [84]. For this reason, the chemical synthesis of marine-derived indolic nitrones has become an interesting research topic. Chemical conversion of indolines into their respective nitrones is still a challenge, as the reaction requires either mild oxidative conditions or a multistep procedure. Alternatively, the enzymatic production of nitrone-functionalized indolines would considerably benefit from its mild reaction conditions and a single high-yield step. Newmister et al. described the single-step biocatalytic conversion of roquefortine C (**47**) and roquefortine C semisynthetic derivatives for the production of corresponding cycloadduct indoline alkaloids (**49**) (Figure 8) [85]. The OxaD enzyme was an indolic nitrone synthase, which was isolated from a marine-derived fungus *Penicillium oxalicum* F30. The desired final products (indoline nitrones) were isolated in pure form with a satisfactory yield after silica gel chromatography. This work illustrated a good example of a biocatalytic synthesis of indoline nitrones, offering a number of applications in chemistry and chemical biology.

Hapalindole-type indole alkaloids are a diverse class of marine natural products with widespread use in modern (agro)chemical, pharmaceutical, and fine chemical industries. The presence of a halogen substituent in the hapalindole-type indole alkaloids plays a vital role in its biological activity, electronic properties, and metabolic stability [86]. Therefore, the development of a synthetic transformation for hapalindole-type alkaloids has been a long-sought goal. The generation of aliphatic carbon–halogen bonds in natural alkaloids requires chemo-, regio-, and stereoselective control of the inert sp^3^-carbon centers. As a result, the chemical synthesis of hapalindole-type indole alkaloids has not yet been achieved. Remarkably, aliphatic halogenases are promising biocatalysts for the halogenation of inactivated aliphatic carbons [87]. Zhu et al. described the application of standalone aliphatic halogenases to introduce bromine instead of chlorine substituents into the family of natural products. The Fe/2OG halogenase WelO5 could convert 12-*epi*-fischerindole U (**50**) and 12-*epi*-hapalindole C (**52**) to 12-*epi*-fischerindole G (**51**) and 12-*epi*-hapalindole E (**53**) with a very good yield in a single step (Figure 9) [88]. The incorporation of a halogen substituent at the C-13 of **50** afforded the corresponding hapalindole-type indole alkaloid (**51**), resulting in significantly enhanced antibiotic activity toward human bacterial and fungal pathogens. This work highlighted the versatility of WelO5 as a halogenase for the production of hapalindole-type indole alkaloids, as well as expanded the application of WelO5 as a biocatalyst and a useful tool for modern medicinal chemistry.

Prenylation reactions catalyzed by prenyltransferases play crucial roles in controlling biomolecular activity, including indole alkaloids. Different enzymes have been used to catalyze the prenylation of different positions in the indole ring. Prenylation reactions can be divided into two types, namely, normal prenylation and reverse prenylation [89], where normal prenylation indicates that prenylation occurs at the C-1 primary center, while reverse prenylation occurs at the C-3 tertiary center. Mori et al. described the chemoenzymatic synthesis of indole alkaloids lyngbyatoxin A and pendolmycin by using indole prenyltransferases TleC and MpnD, respectively (Figure 10) [90]. TleC from *Streptomyces blastmyceticus* was a 42 kDa protein consisting of 391 amino acids, while MpnD from the deep sea *Marinactinospora thermotolerans* was a protein that shared 38% identity with TleC. Both TleC and MpnD catalyzed the reverse prenylation reactions of the cyclic dipeptide (−)-indolactamV (**54**) at the C-7 position of the indole ring, affording lyngbyatoxin A (**55**) and pendolmycin (**57**), respectively. The substrate specificities of TleC and MpnD were also investigated by incubating the purified enzymes with several indole derivatives and prenyl pyrophosphates of various chain lengths. The experimental results illustrated that TleC and MpnD had relaxed substrate specificity toward the prenyl donors, but strict substrate specificity toward the prenyl acceptors. Therefore, a series of unnatural indolactam derivatives with improved/modified biological activity could be produced by TleC and MpnD using the same reverse prenylation strategy.

Selective C–H functionalization is an important transformation in synthetic chemistry, as it does not require additional synthetic handles to oxidize intermediates. Nevertheless, chemical C–H oxidation is still challenging as a result of uncontrolled regioselectivity; thus, its application in the chemical synthesis of biologically active alkaloids is limited [91]. For example, the chemical synthesis of marine-derived norditerpenoid alkaloid nigelladines A−C (Figure 3) requires particular attention to the viability of late-stage C–H oxidation. Alternatively, biocatalytic C–H oxidation has shown excellent regioselectivity and reactivity. Despite this, biocatalytic C–H oxidation has rarely been used for the total synthesis of natural products due to its typically narrow substrate scope [92]. Remarkably, Loskot et al. described the enantioselective chemoenzymatic synthesis of norditerpenoid alkaloid nigelladine A for the first time in 2017 [93]. Total synthesis relied on the late-stage C–H oxidation of the advanced intermediate catalyzed by an engineered cytochrome P450 enzyme. The enzyme successfully catalyzed sit-selective 2° allylic oxidation in the presence of four oxidizable positions. Because selective P450-catalyzed C–H oxidation was a key step, the total synthesis of nigelladine A was accomplished in an expedient 12 steps and with an overall yield of 5%. This work showed that enzymatic transformation could be a significant alternative to the traditional chemical method, especially for late-stage total synthesis of natural products.

As discussed above, prenylated indole alkaloids comprise a large class of marine-derived natural products. Roquefortine C (Figure 4) and its derived alkaloids represent a special type of prenylated indole alkaloids, featuring a unique triazaspirocyclic skeleton. Meleagrin and oxaline are frequently isolated from marine-derived fungus strains, producing roquefortine C-derived alkaloids [94]. Meleagrin contains only one methyl group at the hydroxylamine oxygen, while oxaline is methylated at both the hydroxylamine and the enol oxygens. Roquefortine C-derived alkaloids commonly exhibit neurotoxic and antimicrobial activity. For example, oxaline has shown to be a promising inhibitor against tubulin polymerization with an IC_50_ value of 8.7 μM, while Meleagrin has shown to be an active inhibitor of the bacterial FabI target with IC_50_ values ranging from 1.8 to 6.7 μM [95]. For this reason, synthetic strategies toward roquefortine C-derived alkaloids have become urgent. However, they are still challenging, with respect to the viability of biological activity, which differs within different methylation patterns. Two homologous enzymes OxaG and RoqN have been identified in genome sequences using bioinformatic approaches, which have shown to catalyze penultimate hydroxylamine *O*-methylation and generate oxaline and meleagrin in vitro, respectively [96]. The biochemical activity and crystal structures of the two methyltransferases were characterized in detail, providing structural and mechanistic insights into the enzymatic hydroxylamine methylation of oxaline and meleagrin. This work could be used to further guide efforts toward the chemoenzymatic synthesis of related alkaloids with OxaG/RoqN-catalyzed site-specific, late-stage methylation as the key step.

Psilocybin belongs to a class of indole alkaloids, possessing a unique 4-phosphoryloxy group. Psilocybin is of pharmaceutical interest, as it has been used to treat end-of-life anxiety and depression in advanced clinical trials [97]. Significant progress has been achieved in the chemical synthesis of psilocybin. For example, Yamada et al. successfully synthesized various derivatives containing 6-phosphoryloxy and 6-hydroxyisomers of psilocybin [98]. However, the synthesis of 6-methylated psilocybin is still beyond the capability of existing approaches. A comparison between 6-methylated psilocybin and psilocybin bioactivity showed that no significant differences were observed in the treatment of end-of-life anxiety and therapy refractory depression. Therefore, Fricke et al. described the enzymatic synthesis of 6-methylated psilocybin, to provide a strategy for gram-scale production and to provide sufficient psilocybin in vitro, specifically for comparative in vivo assays [99]. 4-Hydroxy-6-methylindole (**59**) was used as the starting material and was converted to 4-hydroxy-6-methyl-L-tryptophan (**60**) using tryptophan synthase *P. cubensis* TrpB (Figure 11). Subsequently, compound (**60**) was used as the co-substrate for a one-pot reaction using recombinant enzymes PsiM, PsiD, and PsiK, which catalyzed *N*-methylation (6-methylbaeocystin **63**), decarboxylation (4-hydroxy-6methyltryptamine **61**), and phosphorylation (6-methylnorbaeocystin **62**), respectively. In addition, 6-methylated psilocybin (**64**) was obtained in two enzymatic steps with 75% yield. This work provided a completely new strategy for the synthesis of alkaloids and their derivatives.

Tetrahydroprotoberberine and protoberberine are unique alkaloids from marine-derived natural products, as they possess a methyl group at the C-13 position. Notably, 13-methyltetrahydroprotoberberine (13-Me-THPB) and 13-methylprotoberberine (13-Me-PB) alkaloids are biologically active [100]. For example, they were shown to be dopamine D-1 receptor agonists and exhibit anti-hepatitis B and inhibitory activity against reverse transcriptase and enterovirus 71. Isolation from marine sources has typically resulted in low production and separation issues, with regard to structurally closed metabolites. Significant progress has been made in the chemical synthesis of 13-Me THPB and 13-Me-PB alkaloids, although most cases can afford the final products in racemic form. For example, Zhou et al. described the asymmetric synthesis of 12 natural 13-Me-THPBs (**58**) in three steps, with an overall yield of 47–65% [101]. This approach used a chiral ligand (an enantiopure PINAP ligand) to accomplish the stereoselective synthesis of 13-Me-THPBs with 91–96% *ee* (Figure 12a). Roddan et al. demonstrated chemoenzymatic cascades toward 13-Me-THPB and 13-Me-PB using stereoselective Pictet–Spenglerase, regioselective catechol *O*-methyltransferases, and selective chemical Pictet–Spengler reactions (Figure 12b) [102]. This approach had notable advantages, including subsequent reactions and no requirements for workup isolation of any synthetic intermediates. The 13-Me-THPB and 13-Me-PB alkaloids were obtained with 99% yield and 96% *ee*. Remarkably, the obtained 13-Me-THPB alkaloids had a (−)-configuration, which was novel compared to the naturally occurring (+)-isomeralkaloids.

Quinolizinones are important marine-derived alkaloids due to their interesting bioactivity. For example, they are active inhibitors of HIV integrase and phosphoinositide-3-kinase (PI3K), and they exhibit anti-ulcerative and antiallergic activity [103]. For this reason, the development of a synthetic strategy for quinolizinones has been interesting. Numerous studies concerning the chemical synthesis of quinolizinones have been reported; however, most are not cost-effective or multistep, and they require harsh reaction conditions [104,105]. Wang et al. synthesized representative quinolizinones, denoted as 2-hydroxy-*4H*-quinolizin-4-one, scaffolds by integrating three enzymes (Figure 13) [106]. PcPCL (a phenylacetate-CoA ligase from an endophytic fungus *Penicillium chrysogenum*), which catalyzed the conversion of 2-(pyridine-2-yl)acetic acids (**73**) to the corresponding CoA thioester (**74**). In addition, AtMatB (a malonyl-CoA synthase from *Arabidopsis thaliana*) catalyzed the conversion of malonic acids (**75**) into malonyl-CoAs (**76**). Subsequently, 2-pyridylacetyl-CoA (**74**) and malonyl-CoA (**76**) underwent condensation, generating a diketide intermediate by a type III PKS. This intermediate (**77**) suffered from intramolecular cyclization, producing quinolizinone scaffolds (**78**–**82**). This work may provide an entirely new approach to the enzymatic synthesis of pharmaceutically important alkaloids.

## 4. Conclusions and Outlook

Interest in natural products has been mainly spurred by their biological and pharmaceutical activity, along with the fact that approximately half of the drugs approved by the Food and Drug Administration (FDA) originate from natural products. Alkaloids are a large and structurally diverse group of important nitrogen-containing natural products, with a wide variety of pharmacological applications, which many of these compounds exhibit. As discussed in this review, many alkaloids possess antitussive, antimicrobial, and antispasmodic activity, and they have shown promising applications in treating cancer, malaria, and many other diseases. For these reasons, accessing natural alkaloids has been particularly important. However, this process still remains a challenge. Although most marine-derived alkaloids are isolated from marine sources, they still suffer from poor production and/or very expensive and time-consuming separation issues. Total synthesis represents a typical example of modern marine-derived alkaloid production, which may be commercially viable in some cases. However, the chemical synthesis of many alkaloids typically involves multiple steps and environmentally harmful reactions, resulting in unattainable challenges and/or low economic efficiency for more complex products.

Among the employed production approaches, (chemo)enzymatic synthesis has been shown to be a highly valuable, viable, and sustainable synthetic method for marine-derived alkaloid production. Compared to traditional organic synthetic methods, biotechnological methods have several advantages, including mild reaction conditions, excellent enantioselectivity and stereoselectivity, non-hazardous reagents, and fewer toxic byproducts. Indeed, biotechnology methods have already been used to produce marine-derived alkaloids on large scale, for example, in the production of thebaine, a precursor to prescription opioids. The in vivo fermentation process plays an important role in current marine-derived alkaloids and involves significant bioengineering efforts; however, in some cases, poor enantiopurity still needs to be resolved. Alternatively, biotransformation by mimicking biosynthesis in vitro using recombinantly expressed enzymes (enzymatic synthesis) may be a promising approach for marine-derived production. The reviewed research showed the progress of (chemo)enzymatic synthesis for a series of marine-derived alkaloids and the pros and cons of their application on an industrial scale. Admittedly, the use of (chemo)enzymatic synthesis for marine-derived alkaloid production may be limited due to poor enzyme stability and strict substrate specificity. In particular, cascade processes involving a combination of traditional synthetic methods and biocatalytic enzymes would be highly appreciated, especially for the production of complex products using a simpler system than solely enzymatic or organic synthetic routes.

Currently, yields for marine-derived alkaloid production via (chemo)enzymatic approaches have not exceeded a few mg/L, indicating the additional efforts that are needed to optimize this process, including improved enzyme activity and reaction conditions. The close and effective communication and collaboration between organic chemists and enzymologists would definitely enhance the application of enzymatic reactions in alkaloid production. Most of the biotechnological methods discussed in this review have not been applied on an industrial scale. To demonstrate the preparative-scale synthesis of marine-derived alkaloids using enzymatic reactions, additional research on enzyme immobilization, novel screening methods, and enzyme mutagenesis is needed.

## Data Availability

Not applicable.

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
