# Peer review of "Recent Advances in the Synthesis of Marine-Derived Alkaloids via Enzymatic Reactions"

_marinedrugs, 2022, doi:10.3390/md20060368_

Round 1

Reviewer 1 Report

This manuscript of review, submitted to Marine Drugs, describes recent progresses of emzymatic reaction employed in alkaloid syntheses. Topics treated in this manuscript are interesting and useful for readers of Marine Drugs. However, this submitted manuscript is highly reader-unfriendly especially in schemes and figures because of the following reasons. These must be first improved for accepting.

  1. Lacks of uniformity among schemes and figures.
  2. Put compound numbers in main text.
  3. Scheme or Figure must be put if mentioned in the main text.
  4. This referee feels some normal chemical reactions mentioned in schemes are not necessary, for examples, schemes 1, 2, 9 or 13. Please consider.
  5. Resolution of some schemes is not good, for example schemes 1.
  6. TOO MANY MISTAKES in schemes. Recheck very carefully.
  7. Check all references again because of Lacks of uniformity.

Author Response

Dear Reviewer,

We would like to thank you for the valuable revision of the manuscript. The suggestions were considered, and now the revised manuscript includes the proposed improvements.

Please find the responses to the reviewer's comments (marked in red).

Best regards,

Fayene Zeferino Ribeiro de Souza

(on behalf of all authors)

Reviewer 2 Report

This review manuscript reports on the synthesis of marine-derived alkaloids via enzymatic reactions. Recently, much attention has been paid to chemoenzymatic strategies for the synthesis of complex alkaloid natural products because of their unparalleled capability of rapid generation of complex structures. In this regard, the subject of this manuscript appears timely and would attract the attention of Marine Drugs readers. I recommend publication of this manuscript in Marine Drugs provided the authors address the following points.

1) Examples shown in Scheme 2 and Scheme 9 are merely classical applications of enzymes for the desymmetrization of meso compounds, and therefore should be removed from this manuscript.

2) Critical errors in the chemical structure of important compounds:

Scheme 3, the stereochemistry is missing for compounds 28 and 29.

Scheme 8, the nitrile group is missing for compound 56.

Scheme 13, the carbonyl group is missing for compound 89.

The authors should carefully check all the graphical materials.

3) Language polishing is required.

Author Response

(The authors gave the same response as above.)

Reviewer 3 Report

This review would have been an interesting addition to the literature regarding the synthesis of marine-derived alkaloids. However,

1) the authors do not clearly state what distinguishes this review from previous related reviews, which although mentioned are not clearly referenced at the appropriate position in the text.

2) the text needs EXTENSIVE editing of English language and style. It often becomes difficult to comprehent.

3) the rational behid the particular presentation order is not clear and appears to be random.

4) the graphics are inconsistent and need a more careful proof reading from the authors.

Some indicative corrections and recommendations can be found in the attached file.

Author Response

(The authors gave the same response as above.)

Reviewer 4 Report

This review article summarises the use the enzymatic reactions in the synthesis of marine-derived alkaloids.  This topics is rarely reviewed in the literature so it is justifiable that the authors would like to contribute a review in this area.  I don't think there is any problem with the reported scientific content but there are quite a few amendments that need to be made to this manuscript.

1) On page 2, line 93, change "areyearly" to "are yearly" and "publishedin" to "published in".  Line 96 change "2021.." to "2021.".

2) Page 3, line 106, change "ethinoderms" to "echinoderms".  Line 111, change "et al.showed" to "et al. showed".  Line 117, change "IC50values" to "IC50 values".  Line 127, change "N-acetylcysteamine" to "N-acetylcysteamine".  Line 135 change "Thehydrolysis" to "The hydrolysis".

3) Page 5, line 163, change "(S)-Norcoclaurine(9)" to "(S)-Norcoclaurine (9)".  Line 175, add a reference to "oxidative deacrboxylation".  Line 179 " a value of 93%", what exactly is this numerical value refers to?

4) Page 9, line 269, change "isolatenitrile" to "isolate nitrile".

5) Page 10, line 269 change "spongePetrosia" to "sponge Petrosia".  Also, there is a lot of text is dedicated to Petrosins but no diagrams of these natural products are available.  The authors should included that in the manuscript.

6) Page 11, line 355, it is wrong to describe compound 39 as nitrone.

7) Page 12, line 377, change "40afforded" to "40 afforded".  In scheme 8, both compounds (42) and (43) are not properly drawn.

8) Page 13, line 396 change "identitywith" to "identity with".  Line 406, change "Scheme9" to "Scheme 9".  In scheme 9, the structure of compound 48 is wrong.  Also add a diagram for nigelladines A-C.

9) Page 14, add a diagram for roupefortine C.

10) Page 15, line 478, change "12" to "twelve".  Line 480, the phrase "to accomplish stereochemistry" is badly written.  Line 482, change "stereo-selective" to "stereoselective".

11) Page 16, scheme 11, the structure of compound (57) should be straight.

12) Page 18, line 587, change "Mar. Drug" to "Marine Drug".  Ref. 10 does not appear to exist.

I think this manuscript is littered with mistakes and should be amended before reconsideration.

Author Response

Response to Reviewer 4 Comments

This review article summarises the use the enzymatic reactions in the synthesis of marine-derived alkaloids.  This topics is rarely reviewed in the literature so it is justifiable that the authors would like to contribute a review in this area.  I don't think there is any problem with the reported scientific content but there are quite a few amendments that need to be made to this manuscript.

We thank the reviewer for sharing the enthusiasm with us.

Point 1:  On page 2, line 93, change "areyearly" to "are yearly" and "publishedin" to "published in".  Line 96 change "2021.." to "2021.".

Response 1: Revised as required. Please, see the new version.

Point 2: Page 3, line 106, change "ethinoderms" to "echinoderms".  Line 111, change "et al.showed" to "et al. showed".  Line 117, change "IC50values" to "IC50 values".  Line 127, change "N-acetylcysteamine" to "N-acetylcysteamine".  Line 135 change "Thehydrolysis" to "The hydrolysis".

Response 2: Revised as required. Please, see the new version.

Point 3: Page 5, line 163, change "(S)-Norcoclaurine(9)" to "(S)-Norcoclaurine (9)".  Line 175, add a reference to "oxidative deacrboxylation".  Line 179 " a value of 93%", what exactly is this numerical value refers to?

Response 3: Revised as required. Please, see the new version.

                    The added ref is 56. Trisrivirat, D.; Sutthaphirom, C.; Pimviriyakul, P.; Chaiyen, P. Dual activities of oxidation and oxidative decarboxylation by flavoenzymes. ChemBioChem 2022, doi: 10.1002/cbic.202100666.

                  “ a value of 93%” has been changed into “a ee value of 93%”.

Point 4: Page 9, line 269, change "isolatenitrile" to "isolate nitrile".

Response 4: Revised as required. Please, see the new version.

Point 5: Page 10, line 269 change "spongePetrosia" to "sponge Petrosia".  Also, there is a lot of text is dedicated to Petrosins but no diagrams of these natural products are available.  The authors should included that in the manuscript.

Response 5: Revised as required. Please, see the new version.

                     Scheme 7 has been added to the revised manuscript as shown below.

Scheme 7. The chemoenzymatic preparation of the bisquinolizidine alkaloid, (−)- and (+)-petrosin, with the lipase me-diated desymmetrization of 1,3-diol as the key step.

Point 6: Page 11, line 355, it is wrong to describe compound 39 as nitrone.

Response 6: Revised as required. Please, see the new version.

Point 7: Page 12, line 377, change "40afforded" to "40 afforded".  In scheme 8, both compounds (42) and (43) are not properly drawn.

Response 7: Revised as required. Please, see the new version.

                     The correct one is shown in the revised manuscript.

Point 8: Page 13, line 396 change "identitywith" to "identity with".  Line 406, change "Scheme9" to "Scheme 9".  In scheme 9, the structure of compound 48 is wrong.  Also add a diagram for nigelladines A-C.

Response 8: Revised as required. Please, see the new version.

The diagram of nigelladine A-C has been added in the revised manuscript.

Figure 3. The structure of nigelladine A-C.

Point 9: Page 14, add a diagram for roupefortine C.

Response 9: Revised as required. The structure of roupefortine C was added in the revised manuscript.

Figure 4. The structure of roupefortine C.

Point 10: Page 15, line 478, change "12" to "twelve".  Line 480, the phrase "to accomplish stereochemistry" is badly written.  Line 482, change "stereo-selective" to "stereoselective".

Response 10: Revised as required. The phrase "to accomplish stereochemistry” has been changed into “This approach used a chiral ligand (an enantiopure PINAP ligand) to accomplish the stereoselective synthesis of 13-Me-THPBs, with 91–96% ee”.

Point 11: Page 16, scheme 11, the structure of compound (57) should be straight.

Response 11: Revised as required. Please, see the new version.

Point 12: Page 18, line 587, change "Mar. Drug" to "Marine Drug".  Ref. 10 does not appear to exist.

Response 12: Revised as required. Please, see the new version.

I think this manuscript is littered with mistakes and should be amended before reconsideration.

We apologize for that. We have tried our best to improve the revised manuscript following your suggestions. Thank you very much for your helpful criticism and comments.

Round 2

Reviewer 1 Report

please check attached file

Author Response

Dear Reviewer,

First of all, we would like to thank you for the valuable revision of the manuscript and also, we would like to thank you for the efforts and helpful criticism and comments to improve the quality of this manuscript! In essence, we have completely implemented all changes/corrections requested by the reviewer. The same is true for the comments and requirements by the editorial team. A detailed answer to the reviewers’ comments is appended to this letter.

Please find enclosed our revised manuscript ‘Recent advances in the synthesis of marine-derived alkaloids via enzymatic reactions (marinedrugs-1660132)’.

Now the revised manuscript includes the proposed improvements. The revisions are highlighted.

Please find below the responses to the reviewer's comments (marked in red).

Sincerely,

Fayene Zeferino Ribeiro de Souza

(on behalf of all authors)

Reviewer 3 Report

Dear Editors, dear authors,

Taking into account “the certificate of English language editing” provided by the authors in their cover letter, our attention was focused to the graphics of the manuscript.

Unfortunately, in our opinion, the manuscript’s graphics are still not of publication quality.

Their scale and settings vary wildly (e.g., compare the scale of structures in Figure 2 with those in Scheme 3 or Scheme 5; Times font is used for atom labels in Scheme 3 while Arial font is used in Scheme 4). This is not simply a matter of esthetics; at times it leads to profoundly distorted structures (e.g., structure 48 in Scheme 12). More important, they still contain chemical errors (e.g., “KPO4” is mentioned in the reaction conditions in Scheme 5). Furthermore, as it was already pointed out to the authors, wedges (solid or hashed) should be used to indicate absolute stereochemistry while bold bonds (solid or hashed) should be used to indicate relative stereochemistry. Nonetheless, the authors continue to use these stereochemical notations indiscriminately (e.g., structures 3739 in Scheme 7).

In the original manuscript it was stated: “By definition, alkaloids are cyclic molecules with negatively oxidized nitrogen atoms” (page 2, line 96). Although the expression “negatively oxidized nitrogen atoms” was indeed used by the authors of the cited reference 35, we felt it could be confusing since an oxidized nitrogen atom has a positive oxidation number while a reduced nitrogen atom has a negative oxidation number. Thus, it was suggested to rephrase the above sentence to: “By definition, alkaloids are cyclic molecules containing nitrogen atoms with negative oxidation numbers”. In response, the authors revised the text to “By definition, alkaloids are cyclic molecules with positively oxidized nitrogen atoms” (while citing the same reference!). In retrospect, it is preferable to keep the original phrasing.

The authors should make sure that all schemes and figures are numbered in sequential order (currently there are schemes 1-8, two schemes numbered 12, scheme 9 is missing, etc.).

Although overall the manuscript has been improved, based on the above observations it does not warrant publication in Marine Drugs in its present form.

Author Response

(The authors gave the same response as above.)

Reviewer 4 Report

The current version of this manuscript is improved when compares to the previous one.  However, I notice there are two minor mistakes in scheme 7.  Compound (39) should be an alcohol and not a TBS silyl ether.  Also. is MeOH missing as a reagent in the transformation of (42) to (43)?

I think this manuscript is acceptable for publication once these two mistakes are amended.

Author Response

Point 1: The current version of this manuscript is improved when compares to the previous one.  However, I notice there are two minor mistakes in scheme 7.  Compound (39) should be an alcohol and not a TBS silyl ether.  Also. is MeOH missing as a reagent in the transformation of (42) to (43)?

Response 1: Revised as required. Please, see the new version.

Scheme 7 has been changed into the following version:

I think this manuscript is acceptable for publication once these two mistakes are amended.

We would like to thank you once again for the efforts and helpful criticism and comments to improve the quality of this manuscript!
